# Morphological and Molecular Identification of Plant Pathogenic Fungi Associated with Dirty Panicle Disease in Coconuts (*Cocos nucifera*) in Thailand

**DOI:** 10.3390/jof8040335

**Published:** 2022-03-23

**Authors:** Anurag Sunpapao, Nakarin Suwannarach, Jaturong Kumla, Reajina Dumhai, Kanamon Riangwong, Sunisa Sanguansub, Samart Wanchana, Siwaret Arikit

**Affiliations:** 1Agricultural Innovation and Management Division (Pest Management), Faculty of Natural Resources, Prince of Songkla University, Songkhla 90110, Thailand; anurag.su@psu.ac.th; 2Research Center of Microbial Diversity and Sustainable Utilization, Chiang Mai University, Chiang Mai 50200, Thailand; suwan.462@gmail.com (N.S.); jaturong_yai@hotmail.com (J.K.); 3Rice Science Center, Kamphaeng Saen Campus, Kasetsart University, Nakhon Pathom 73140, Thailand; reajina.d06@gmail.com; 4Department of Biotechnology, Faculty of Engineering and Industrial Technology, Sanamchandra Palace Campus, Silpakorn University, Nakhon Pathom 73000, Thailand; kanamonnueng@gmail.com; 5Department of Entomology, Faculty of Agriculture at Kamphaeng Saen, Kamphaeng Saen Campus, Kasetsart University, Nakhon Pathom 73140, Thailand; agrssss@ku.ac.th; 6National Center for Genetic Engineering and Biotechnology (BIOTEC), National Science and Technology Development Agency (NSTDA), Pathum Thani 12120, Thailand; samart.wan@biotec.or.th; 7Department of Agronomy, Faculty of Agriculture at Kamphaeng Saen, Kamphaeng Saen Campus, Kasetsart University, Nakhon Pathom 73140, Thailand

**Keywords:** flower discoloration, fungi, pathogenicity test, morphology, molecular techniques

## Abstract

Dirty panicle disease in coconuts (*Cocos nucifera*) was first observed in the KU-BEDO Coconut BioBank, Nakhon Pathom province, Thailand. The occurrence of the disease covers more than 30% of the total coconut plantation area. The symptoms include small brown to dark brown spots and discoloration of male flowers. Herein, three fungal strains were isolated from infected samples. Based on the morphological characteristics the fungal isolates, they were classified into two genera, namely, *Alternaria* (Al01) and *Fusarium* (FUO01 and FUP01). DNA sequences of internal transcribed spacer (ITS), glyceraldehyde 3-phosphate dehydrogenase (GAPDH), translation elongation factor 1-α (*tef1-α*), and RNA polymerase II second largest subunit (*rpb2*) revealed Al01 as *Alternaria burnsii*, whereas DNA sequences of ITS, *rpb2*, and *tef1-α* identified FUO01 and FUP01 as *Fusarium clavum* and *F. tricinctum*, respectively. A pathogenicity test by the agar plug method demonstrated that these pathogens cause dirty panicle disease similar to that observed in natural infections. To the best of our knowledge, this is the first report on the novel dirty panicle disease in coconuts in Thailand or elsewhere, demonstrating that it is associated with the plant pathogenic fungi *A. burnsii*, *F. clavum*, and *F. tricinctum*.

## 1. Introduction

Coconuts (*Coconut nucifera*) are a member of the palm tree family, which belongs to the family Arecaceae. Coconut plantations are grown in over 90 countries worldwide, especially in tropical areas, with over 12 million hectares, of which over 80% of the production is in Asia [1,2]. Coconut plants are considered multipurpose perennial plantation crops. They provide a nutritious drink, edible nutritious products, edible coconut oil, fiber for commercial value, and coconut shell for fuel and industrial uses. In some developing countries, coconuts serve as a cash crop and copra, being one of the few sources of income for several households [3]. A range of value-added products has been developed from coconuts, resulting in social and economic benefits worldwide.

As coconut plantations are primarily located in tropical areas, an environment that favors pathogen infection and disease spread, coconut trees face several diseases in all stages of growth. Fungal diseases can negatively impact both the quality and quantity of coconut production. For instance, the fungus *Ceratocystis paradoxa* has been documented to cause stem bleeding in coconuts in Hainan, China [4]. The fungus *Pestalotiopsis menezesiana* has been found to cause leaf blight in coconuts [5]. *Lasiodiplodia theobromae* has been also documented as the major fungus causing fruit rot and nut fall in coconuts [6]. Leaf spotting of coconut seedlings has been observed, for the first time, to be caused by *Bipolaris setariae* [7]. Recently, postharvest stem end rot of coconuts was documented in China, and the causal pathogen was identified as *L. theobromae* [8].

In Thailand, coconuts are considered economic plants due to the multifarious uses of all of their parts in the commercial sector. The total cultivation area of coconuts in Thailand is approximately 19,840 hectares overall, with the production of 320,000 tons of coconuts (Ministry of Commerce, Thailand). Two types of coconuts are usually grown: Tall coconuts, for which mature fruits are used, and dwarf coconuts, for which young fruits are used [9]. The major areas of tall coconuts are in the south, including the Chumporn, Prachuap Khiri Khan, and Nakhon Si Thammarat provinces, while the major areas of dwarf coconuts are in central Thailand, including the Ratchaburi, Nakhon Pathom, Samut Sakhon, and Cha Choeng Sao provinces. Approximately 80–90% of coconut products are exported. In particular, the demand for whole young nuts and coconut water increases yearly [10,11]. As Thailand is located in tropical and subtropical areas, the weather favors pathogen germination and disease spread [12,13]. Recently, the plantation of coconuts has faced dirty panicle disease in the KU-BEDO Coconut BioBank, Nakhon Pathom province. According to the previous literature, there is no report on fungi associated with dirty panicle disease in coconuts. Therefore, this research aimed to identify the causal agent of dirty panicle disease in coconuts by its morphology and molecular properties, as well as to test the pathogenicity.

## 2. Materials and Methods

### 2.1. Field Observation, Symptom Recognition, and Sample Collection

Field observation was carried out in the KU-BEDO Coconut BioBank, Kasetsart University, Khampaeng Saen Campus, Nakhon Pathom, Central Thailand (N 14.015619 E 99.958970) (Appendix A). The cultivation area covers approximately 4.8 hectares with 770 coconut trees. Coconuts exhibiting dirty panicle disease were photographed, collected in a sterile plastic bag, kept on ice box, and brought to the laboratory, where experiments were subsequently conducted. In this study, 15 infected panicles were collected for further study.

### 2.2. Fungal Isolation

Fungal isolation was conducted by tissue transplantation according to the method previously described by Daengsuwan et al. [14,15] with some modifications. The infected tissues were cut into small pieces (0.3 × 0.3 cm) and surface-sterilized with 70% ethanol. The infected pieces were then followed by soaking in 10% sodium hypochlorite (NaOCl). The excess NaOCl was removed by washing in sterile distilled water (DW). The pieces of infected samples were dried on sterile Whatman^®^ filter paper in a laminar air flow cabinet. Pieces of infected tissues were placed on 1.5% water agar (WA) and incubated at an ambient temperature (28 ± 2 °C) for three days. Hyphal tips recovered from infected tissues were cut and transferred to potato dextrose agar (PDA) for further study.

### 2.3. Pathogenicity Test

To test which fungal strains isolated from infected samples could cause disease of the panicles of coconuts in a laboratory, the agar plug method was performed following the method described by Pornsuriya et al. [16] and Runagwong et al. [17] with some modifications. The healthy panicles of 12 coconuts were prepared for each strain of inoculation. The fungal isolates were cultured on PDA and incubated at an ambient temperature (28 ± 2 °C) for seven days. The panicles were wounded (three panicles) using fine sterile needles. Mycelial plugs from the seven-day-old culture were cut from PDA plates and directly placed onto the wounded panicles. The inoculated panicles were incubated in a sterile plastic box to maintain humidity for 72 h at 28 ± 2 °C and with a 12 h/12 h light/dark cycle. Three healthy panicles were inoculated with PDA alone and served as the control. The experiments were repeated three times. The progress of symptoms was observed after the first week of inoculation.

### 2.4. Morphological Identification

Fungal isolates that were able to cause dirty panicle disease in coconuts were cultured on PDA for seven days and subjected to macroscopic and microscopic study by using both a stereomicroscope (Leica S8AP0, Leica, Wetzlar, Germany) and a compound microscope (Leica DM750, Leica, Wetzlar, Germany) with 30 replicates (*n* = 30). The growth rate, colony characteristics, and dimensions of the conidia were measured and compared to known species identification. The pure fungal cultures were then deposited in the Culture Collection of Pest Management, Faculty of Natural Resources, Prince of Songkla University.

### 2.5. Molecular Identification 

Pure cultures of each fungal strain were cultured on PDA for three days and subjected to DNA extraction via a DNA Extraction Mini Kit (FAVORGEN, Ping-Tung, Taiwan) following the manufacturer’s protocol. A polymerase chain reaction (PCR) was performed in a 20 μL volume containing 1.0 μL of the DNA template, 1.0 μL of each primer, 10.0 μL of 2X Quick Taq^®^ HS DyeMix (TOYOBO, Saitama, Japan), and 7 μL of deionized water. PCR amplification of the internal transcribed spacer (ITS), part of RNA polymerase 2 (*rpb2*), translation elongation factor 1-alpha (*tef1*), and glyceraldehyde-3-phosphate dehydrogenase (GAPDH) genes were amplified using ITS1/ITS4 [18], fRPB2-5f/fRPB2-7cr [19], Tef1-728F/Tef1-986R [20], and GPD1/GPD2 [21] primer pairs (Appendix A), respectively, in the following thermal conditions: 94 °C for 2 min, followed by 35 cycles of 94 °C for 2 min, annealing at a temperature dependent on the amplified gene (ITS at 50 °C, *rpb2* and *tef1* at 52°C, and GAPDH at 60 °C) for 60 s and 72 °C for 1 min, and a final 72 °C for 10 min in a peqSTAR thermal cycler (PEQLAB Ltd., Fareham, UK). The PCR products were observed on 1% agarose gels stained with ethidium bromide under UV light. The PCR products were purified using a PCR Clean-Up Gel Extraction NucleoSpin^®^ Gel and PCR Clean-Up Kit (Macherey-Nagel, Düren, Germany). The purified PCR products were directly sequenced via Sanger sequencing, carried out by the 1st Base Company (Kembangan, Selangor, Malaysia) using the PCR primers mentioned above. The sequences were used to query the GenBank gene sequence database via BLAST (http://blast.ddbj.nig.ac.jp/top-e.html, accessed on 5 February 2022).

Multiple sequence alignment was performed with MUSCLE [22] and improved where necessary using BioEdit v. 6.0.7 [23]. A phylogenetic tree was constructed using the maximum likelihood (ML) and Bayesian inference (BI) methods. ML analysis was carried out on RAxML v7.0.3 under the GTRCAT model with 25 categories and 1000 bootstrap (BS) replications [24,25] via the online portal CIPRES Science Gateway v. 3.3 [26]. BI analysis was performed with MrBayes v3.2.6 [27]. For the BI analysis, six simultaneous Markov chains were run for one million generations with random initial trees, wherein every 1000 generations were sampled. A burn-in phase was employed to discard the first 2000 of the trees, while the remaining trees were used to construct the 50% majority-rule consensus phylogram with calculated Bayesian posterior probabilities (PPs). Tree topologies were visualized in FigTree v1.4.0 [28].

## 3. Results

### 3.1. Symptom Recognition

The occurrence of dirty panicle disease in the KU-BEDO Coconut BioBank was approximately 30% of the surveyed coconut. The primary symptom was observed on male flowers after emerging from the flower bud with flower discoloration. The symptoms included small dark brown spots (0.1–0.3 cm in diameter), with lesions then distributed throughout the flowers and peduncles, resulting in flower drop (Figure 1).

### 3.2. Pathogenicity Test

A total of 20 fungal isolates were isolated from 15 infected panicles. These 20 isolates were tested for pathogenicity on healthy panicles, and only three isolates were identified to cause dirty panicle disease, similar to that observed in the field (Figure 2). Primary identification based on the colony’s characteristics and the morphology of three isolates resulted in grouping into *Alternaria* sp. (Al01) and *Fusarium* sp. (FUO01 and FUP01). The fugal strains Al01, FUO01, and FUP01 were re-isolated from inoculated panicles, and the morphology again matched *Alternaria* sp. for Al01 and *Fusarium* sp. for FUO01 and FUP01, confirming Koch’s postulates. Therefore, the fungal strains Al01, FUO01, and FUP01 were selected for identification based on their morphology and molecular properties.

### 3.3. Morphological Identification

The fungal strain Al01 colony was light to dark brown, flocculent with entire edges, dark gray in reverse. The culture colony produced chains, multicellular, obclavate to obpyriform conidia, 20.38–4.26 × 9.35–16.98 µm with 3–5 longitudinal and 0–2 transverse septa (Figure 3). The fungal strain FUO01 colony was light yellow to peach color with entire edges (Figure 4). The macroconidia were thick-walled and moderately curved, 3–6 septa, 18.07–41.52 × 3.24–7.37 µm. The microconidia were 0 septa, 2.68–17.26 × 0.2–11.9 µm. The chlamydospores were solitary or in chains. The fungal strain FUP01 colony was pinkish to purple in a concentric ring with entire edges (Figure 4). The macroconidia were thick-walled and moderately curved, 7–8 septa, 30.52–39.54 × 2.78–5.02 µm. The microconidia were abundant, 0–4 septa, 4.26–10.50 × 6.28–19.05 µm. The chlamydospores were abundant, formed into chains or clusters. Three strains were deposited in the Culture Collection of the Pest Management Department, Faculty of Natural Resources, Prince of Songkla University, Thailand, with accession numbers Al01, FUO01, and FUP01.

### 3.4. Molecular Identification

Based on the morphological characteristics, fungal strain Al01 was initially identified as belonging to the genus *Alternaria*, whereas fungal strains FUO01 and FUP01 were initially identified as belonging to the genus *Fusarium*. The fungal identification was then further confirmed by multi-gene phylogenetic analyses. The ITS, GAPDH, *tef1*, and *rpb2* sequences of fungal strain Al01 were deposited under GenBank numbers OM570332, OM630452, OM630454, and OM630453, respectively. Notably, the ITS (OM570333 and OM570550), *rpb2* (OM630455 and OM630457), and *tef1* (OM630456 and OM630458) sequences of fungal strains FUO01 and FUP01, respectively, were deposited in the GenBank database.

For identification of *Alternaria*, the combined sequence of ITS, GAPDH, *tef1*, and *rpb2* were used. This dataset consisted of 18 taxa, and the aligned dataset comprised 2312 characters, including gaps (ITS: 1–729; GAPDH: 730–1314; *tef1*: 1315–1556; and *rpb2*: 1557–2312). ML analysis of the combined dataset yielded a best scoring tree with a final ML optimization likelihood value of –4505.8633. The matrix contained 144 distinct alignment patterns with 7.21% undetermined characters or gaps. The estimated base frequencies were recorded as follows: A = 0.2437, C = 0.2728, G = 0.2456, and T = 0.2376; substitution rates of AC = 0.9723, AG = 2.7124, AT = 0.6971, CG = 0.5980, CT = 7.4776, and GT = 1.0000; and gamma distribution shape parameter alpha = 0.1719. The tree length value was equal to 0.3992. In addition, the final average standard deviation of the split frequencies at the end of the total MCMC generations was calculated to be 0.00853 through BI analysis. Phylograms of the ML and BI analyses were similar in terms of topology (data not shown). The phylogram obtained from the ML analysis is shown in Figure 5. A phylogram successfully assigned the fungal isolate Al01 to the same clade of *A*. *burnsii* containing the type species (CBS 107.38). This clade formed a monophyletic clade with high BS (100%) and PP (1.0) supports.

The combined sequence of ITS, *rpb2* and *tef1* were used to identify the *Fusarium* species. This dataset consisted of 23 taxa, and the aligned dataset was comprised of 2093 characters, including gaps (ITS: 1–612; *rpb2*: 613–1463; and *tef1*: 1464–2093). ML analysis of the combined dataset yielded a best scoring tree with a final ML optimization likelihood value of –5512.3754. The matrix contained 402 distinct alignment patterns with 11.46% undetermined characters or gaps. The estimated base frequencies were recorded as follows: A = 0.2585, C = 0.2883, G = 0.2508, and T = 0.2323; substitution rates of AC = 1.1126, AG = 2.6234, AT = 1.0772, CG = 0.4744, CT = 7.1567, and GT = 1.0000; and gamma distribution shape parameter alpha = 0.4065. The tree length value was equal to 0.4086. In addition, the final average standard deviation of the split frequencies at the end of the total MCMC generations was calculated to be 0.00642 through BI analysis. The phylograms of the ML and BI analyses were similar in terms of topology (data not shown). The phylogram obtained from the ML analysis is presented in Figure 6. Fungal strains FUO01 and FUP01 were assigned to the monophyletic clades of *F. clavum* and *F. tricinctum*, respectively, each clade having high BS (100%) and PP (1.0) supports.

## 4. Discussion

In this study, dirty panicle disease of coconuts was first isolated and described in Thailand. The fungi associated with this disease were characterized into at least two genera of the plant pathogenic fungi *Alternaria* and *Fusarium*. Based on the morphological and molecular properties of multiple DNA sequences, the pathogens were identified as *Alternaria burnsii*, *Fusarium clavum*, and *F. tricinctum*. The three fungal strains are known as plant pathogenic fungi that infect several plant species.

Identification based on morphology is a primary step to classify fungal pathogens at the genus level. Paul et al. [29] described the morphology of the fungal colony, size, and shape (obclavate to pyriform) of *A. burnsii*, a seed-borne pathogen of *Cucurbita maxima* in Bangladesh. Our results are in agreement with previous reports of the morphology of the Al01 colony, being short with a thin conidial size, typical for *A. burnsii*. The morphology of *F. clavum* FUO01 found in this study was similar to that of previous report [30]. Furthermore, *F. tricinctum* FUP01 showed a similar morphology to *F. tricinctum*, causing leaf spots on *Hosta fortunei* in Italy [31], whose colony produced the pigment, size, and shape of macro- and microconidia.

For the identification of plant pathogenic fungi at the species level, the molecular properties of DNA sequences were used in combination with the morphology [14,16]. For instance, Paul et al. [26] used multiple DNA sequences of ITS, small subunit (SSU), glyceraldehyde-3-phosphate dehydrogenase (GAPDH), *gpd*, and *Alternaria* major allergen (*Alt a1*) to identify *A. burnsii* as a seed-borne pathogen of *C. maxima*. Furthermore, Al-Nadabi et al. [32] used DNA sequences of ITS, *gpd*, *rpb2*, and *tef1-α* to identify *A. burnsii* causing leaf spots on wheat and date palms. Our results are in agreement with the previous report mentioned above, with multiple DNA sequences of ITS, *gpd*, *rpb2*, and *tef1-α* successfully identifying Al01 as *A. burnsii*.

To identify *Fusarium* species at the species level, the molecular properties of multiple DNA sequences were also examined. Gilardi et al. [30] used multiple DNA sequences of *rpb2* and *tef1-α* to characterize *F. clavum*, the pathogen of leaf spots and fruit rot in tomatoes. Furthermore, Garibaldi et al. [31] also used multiple DNA sequences of *rpb2* and *tef1-α* to identify *F. tricinctum,* the causal agent of leaf spots on *Hosta fortunei*. Based on the results from our study, multiple DNA sequences of ITS, *rpb2*, and *tef1-α* are able to identify the pathogen causing dirty panicle disease in coconuts as *F. clavum* and *F. tricinctum*.

The fungi of both genera are plant pathogens and are commonly found on the grain surface and in soil and plant tissues [33,34], causing diseases in several plant species and capable of reducing the quality and quantity of plant production worldwide [35,36,37]. *Fusarium* species have been reported to co-infect with some pathogenic fungi; for instance, *F. circinatum* and *Phytophthora* spp. cause pine pitch canker disease on *Pinus radiate* [38]. The members of *Alternaria* and *Fusarium* are associated with the grain discoloration of oats, wheat, and barley [33,39]. In this study, we isolated and characterized the fungal pathogens causing dirty panicle disease in coconut, and we found that at least two genera of plant pathogenic fungi; *Alternaria* and *Fusarium* are associated with this disease. Pathogenicity test and re-isolation from inoculated plants confirmed that fungi in both genera caused the disease. Our results are in agreement with a previous report that fungi in the genus *Alternaria* may co-infect with fungi in the genus *Fusarium* [40]. This finding allowed us to determine that at least two genera of plant pathogenic fungi, *Alternaria* and *Fusarium*, cause dirty panicle disease in coconuts.

The fungi in the genus *Alternaria* consist of diverse species and cause several plant diseases. *Alternaria burnsii* is a small-spore species of the section *Alternaria*, which has been reported to cause gray leaf spots in traditional Chinese medicinal plants [41], cumin (*Cucumin cyminum*) blight [42], pumpkin seed rot in *Cucurbita maxima* [29], and date palm leaf spots (*Phoenix dactylifera*) and wheat (*Triticum aestivum*) [32]. *Fusarium* species have also been reported to cause several diseases in various plant species. *Fusarium clavum* is considered a pathogen that causes leaf disease in vegetable crops [43], leaf spots and fruit rot in tomatoes [30], and brown spots on rose petals [44]. Meanwhile, *F. tricinctum* causes head blight in wheat [45] and wilting of the branches and leaves of apple trees [46]. However, according to the USDA database, there are no reports of the three fungi causing diseases in coconuts in Thailand or elsewhere. Therefore, to the best of our knowledge, this is the first report of *Alternaria burnsii*, *F. clavum*, and *F. tricinctum* causing dirty panicle disease in coconuts.

## 5. Conclusions

Dirty panicle disease in coconut plants, caused by at least two genera of plant pathogenic fungi (*Alternaria* and *Fusarium*), was first described in Thailand. The fungi were isolated from infected plants and identified based on the morphological characteristics and molecular properties of multiple DNA sequences. A pathogenicity test revealed similar symptoms under in vivo conditions to those observed in the field. This is the first report of *Alternaria burnsii*, *F. clavum*, and *F. tricinctum* causing dirty panicle disease in coconuts in Thailand and elsewhere. Further study on disease management is needed to verify this in future.

## Figures and Tables

**Figure 1 jof-08-00335-f001:**
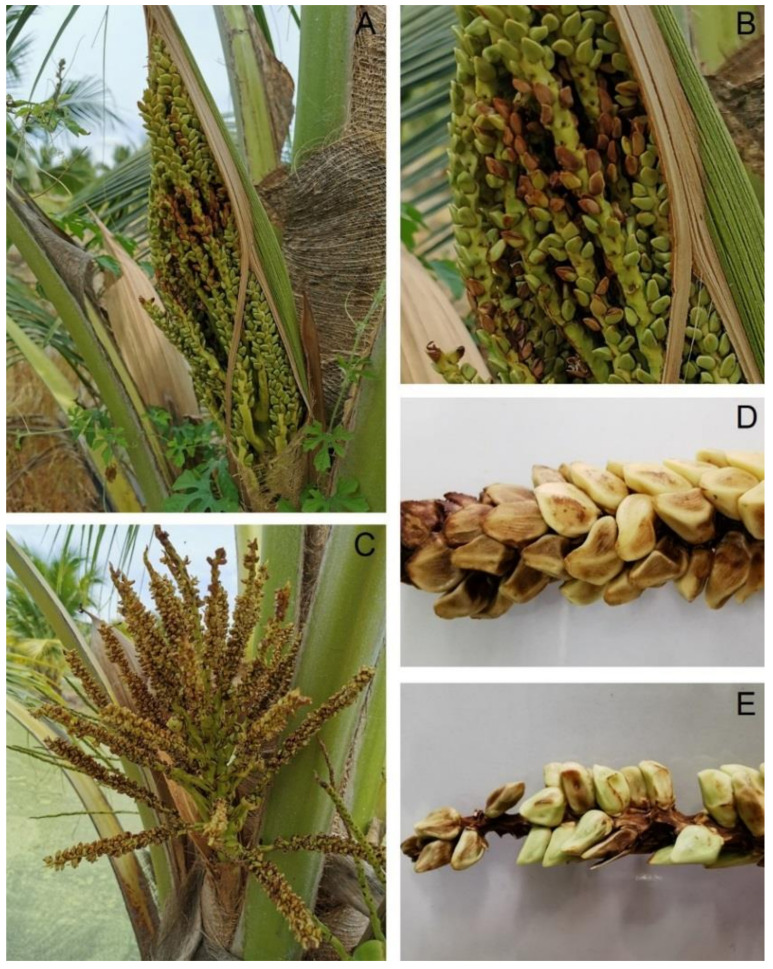
Dirty panicle disease observed on coconuts in KU-BEDO Coconut BioBank: Discoloration of male flowers (**A**) and zoomed-in view of infected flowers (**B**), flower drop (**C**), and infected panicles (**D**,**E**).

**Figure 2 jof-08-00335-f002:**
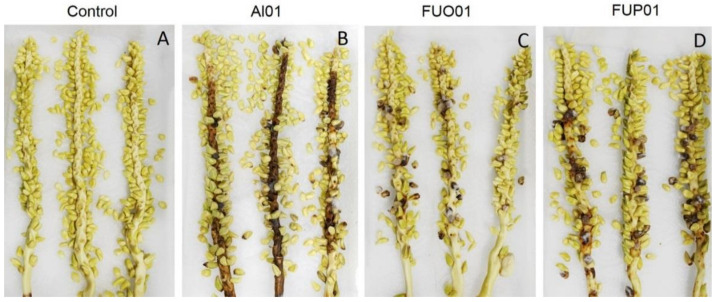
Pathogenicity test of *Alternaria* sp. and *Fusarium* sp. on coconut panicles: Control group (**A**), coconut panicles inoculated with *Alternaria* sp. Al01 (**B**), *Fusarium* sp. FUO01 (**C**), and *Fusarium* sp. FUP01 (**D**).

**Figure 3 jof-08-00335-f003:**
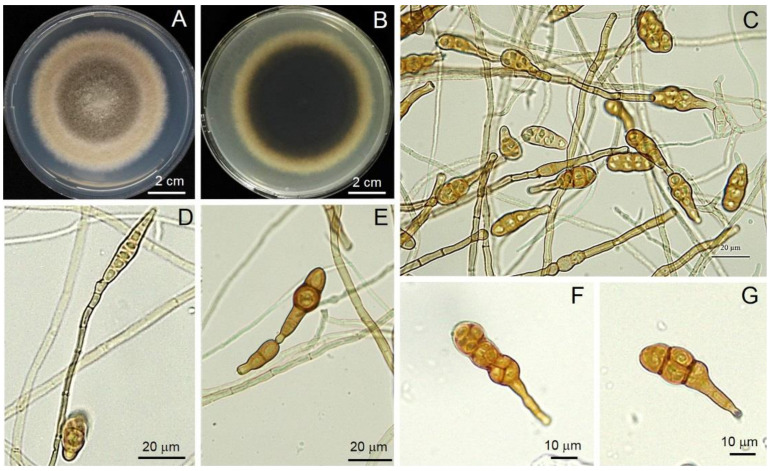
General morphology of *Alternaria* sp. (Al01): Colony on PDA from the top (**A**) and bottom view (**B**); hyphae and diverse shape of conidia (**C**–**G**).

**Figure 4 jof-08-00335-f004:**
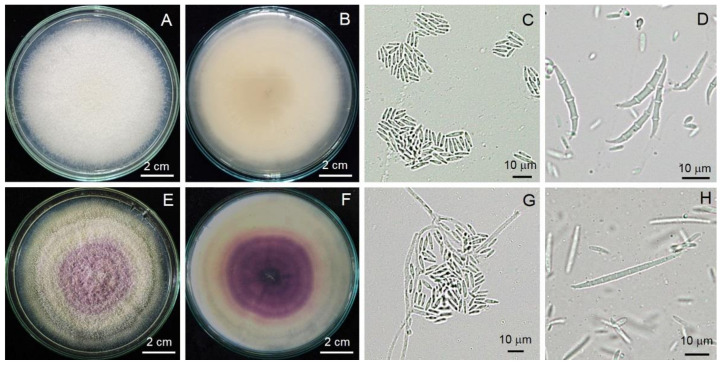
General morphology of *Fusarium* sp. FUO01 (**A**–**D**) and FUP01 (**E**–**H**): Colony on PDA from the top (**A**,**E**) and bottom view (**B**,**F**); hyphae and diverse shape of conidia (**C**,**D**,**G**,**H**).

**Figure 5 jof-08-00335-f005:**
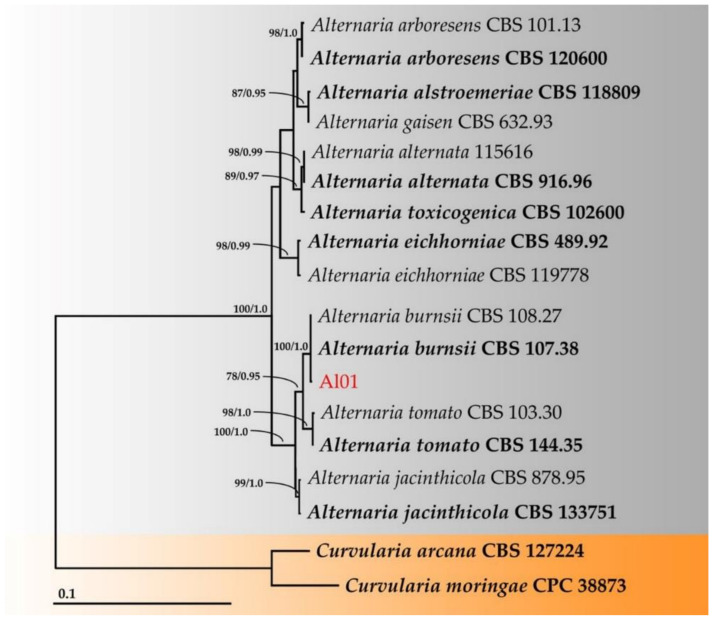
Phylogram derived from the maximum likelihood analysis of 18 taxa of the combined ITS, GAPDH, *tef1*, and *rpb2* sequences. *Curvularia arcana* CBS 127224 and *C. moringae* CPC 38873 were used as the outgroups. The numbers above the branches represent bootstrap percentages (**left**) and Bayesian posterior probabilities (**right**). Bootstrap values ≥75% and Bayesian posterior probabilities ≥0.90 are shown. The scale bar represents the expected number of nucleotide substitutions per site. The sequence of the fungal species obtained in this study is in red. The type species are in bold.

**Figure 6 jof-08-00335-f006:**
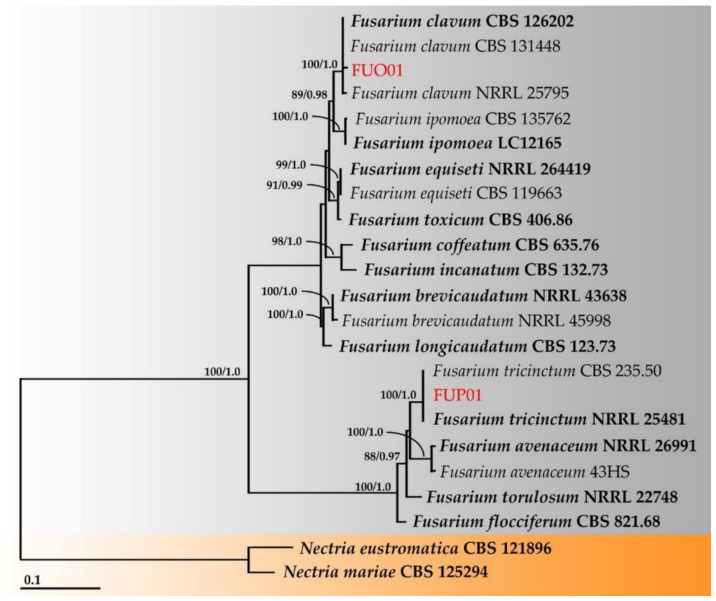
Phylogram derived from the maximum likelihood analysis of 23 taxa of the combined ITS, *rpb2*, and *tef1* sequences. *Nectria eustromatica* CBS 121896 and *N. mariae* CBS 125294 were used as the outgroups. The numbers above the branches represent bootstrap percentages (**left**) and Bayesian posterior probabilities (**right**). Bootstrap values ≥75% and Bayesian posterior probabilities ≥0.90 are shown. The scale bar represents the expected number of nucleotide substitutions per site. The sequences of the fungal species obtained in this study are in red. The type species are in bold.

## Data Availability

The DNA sequence data obtained in this study were deposited in GenBank with accession numbers for ITS (OM570332), GAPDH (OM630452), *tef1* (OM630454), and *rpb2* (OM630453) of Al01 and ITS (OM570333, OM570550), *rpb2* (OM630455, OM630457), and *tef1* (OM630456, OM630458) sequences of fungal strains FUO01 and FUP01, respectively.

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
