# Peer review of "Morphological and Molecular Identification of Plant Pathogenic Fungi Associated with Dirty Panicle Disease in Coconuts (*Cocos nucifera*) in Thailand"

_jof, 2022, doi:10.3390/jof8040335_

Round 1
Reviewer 1 Report
Dear authors,
Article is well written and describe the disease and their identifications with suitable techniques. The reported disease is really very important to identify and manage as it affect the productivity of the crop.
Comments: As the disease reported for the first time. I would like to request you to add the figure of field, characterization gel figures for ITS, gene specific ITS primers amplifications pics in supplementary file of the manuscript for more proof of your work.
Author Response
Comments: As the disease reported for the first time. I would like to request you to add the figure of field, characterization gel figures for ITS, gene specific ITS primers amplifications pics in supplementary file of the manuscript for more proof of your work.
Response:
We would like to thank the reviewer for the suggestion. We have now provided the figure of KU-BEDO Coconut Biobank field and an example of the disease occurring in the field in Supplementary Figure S1 of the revised manuscript. We are unable to provide the gel pictures for ITS and other genes as we did not perform the molecular characterization based on the polymorphisms of the ITS and gene-specific PCR products on the gels. Instead, we sequenced all the PCR products of these genes and used the sequences for the phylogenetic tree analysis. All the sequences have been submitted to the NCBI for ITS (accession number OM570332), GAPDH (OM630452), tef1 (OM630454), and rpb2 (OM630453) of Al01 and ITS (OM570333, OM570550), rpb2 (OM630455, OM630457), and tef1 (OM630456, OM630458) sequences of fungal strains FUO01 and FUP01, respectively. The primer sequences and references were also provided in Supplementary Table S1 in the revised version of the manuscript.
Reviewer 2 Report
This is an interesting paper about the identification, for the first time, of the plant pathogenic fungi Alternaria burnsii, Fusarium clavum, and F. tricinctum as the novel causal agents of dirty panicle disease of coconuts in Thailand or elsewhere based on the morphological characteristics and molecular properties (ITS, GAPDH, tef1, and rpb2). The manuscript shows interesting and novel results, and the objectives are very clear. I would therefore recommend accepting this manuscript.
Author Response
We would like to thank the reviewer for taking the valuable time to review our work. The academic English used in this manuscript was also proofed by the MDPI English Editing Service.
Reviewer 3 Report
I consider it is a valuable study because it detects the probable fungal causal agents of a disease in coconuts.
I make the following suggestions to the authors
Line 69-70: “Recently, the plantation of coconuts has faced dirty panicle disease in the KU-BEDO Coconut BioBank, Nakhon Pathom province. “…Could you provide a report or published scientific work with some statistics and that can be cited?
Line 80 . collected in a plastic bag, … I suggest writing as follows “collected in a sterile plastic bag”
Line 101 and 102… The inoculated panicles were incubated in a plastic box… I suggest writing as follows “The inoculated panicles were incubated in a sterile plastic box”
Line 112… The fungal cultures were then deposited in the Culture…. I suggest writing as follows “The pure fungal cultures were then deposited in the Culture…
Line 183-184 “Three strains were deposited in the Culture Collec-183 tion of the Pest Management Department, Faculty of Natural Resources, Prince of Songkla 184 University, Thailand, with accession numbers Al01, FUO01, and FUP01.” I recommend that the three characterized strains be deposited in a more visible and accessible fungal collection, perhaps in WFCC affiliated culture collections.
Line 288 -289 Our results are in agreement with a previous report that fungi in the genus Alternaria may coinfect with fungi in the genus Fusarium. I suggest clarifying whether the isolated strains come from the same infected panicle.
Line 311 “in vivo” should be written in italics
